# UK Dog Owners’ Pre-Acquisition Information- and Advice-Seeking: A Mixed Methods Study

**DOI:** 10.3390/ani14071033

**Published:** 2024-03-28

**Authors:** Rebecca Mead, Katrina E. Holland, Rachel A. Casey, Melissa M. Upjohn, Robert M. Christley

**Affiliations:** Dogs Trust, 17 Wakley Street, London EC1V 7RQ, UK; katrina.holland@dogstrust.org.uk (K.E.H.); rachel.casey@dogstrust.org.uk (R.A.C.); midge.upjohn@gmail.com (M.M.U.); robert.christley@dogstrust.org.uk (R.M.C.)

**Keywords:** dogs, dog acquisition, pre-acquisition research, pre-acquisition behaviours, preparatory research

## Abstract

**Simple Summary:**

Little is known about the information or advice that prospective owners seek prior to acquiring a dog. This paper reports findings from a mixed methods study into dog acquisition in the UK and focuses on a subset of owners who undertook research prior to acquiring their dog. Those who sought advice or information most often consulted websites, friends and family, and online forums. Topics researched included general information about dogs, information about breeds or types of dogs, owner requirements and dog suitability, aspects of dog ownership, and how to source a dog. Most prospective owners who conducted research reported finding all the information they wanted but some found conflicting advice and did not know which sources to trust. The findings will be of interest to those who provide advice related to dog acquisition and dog ownership, in order to improve the knowledge and decisions made by prospective owners.

**Abstract:**

Dogs are the most common pet animal species in the UK. Little is known about information and advice gathering within the process of dog acquisition, nor what pre-acquisition research encompasses. This study aimed to better understand the preparatory research undertaken by prospective dog owners in the UK. A 2019 online survey collected quantitative and qualitative data about dog acquisition. Analyses within this study focused on a subset of these current (*n* = 4381) and potential (*n* = 2350) owners who had undertaken research, or who reported planning to do so, before acquiring a dog. Additional qualitative data were collected through semi-structured interviews with current (*n* = 24) and potential (*n* = 8) dog owners. Among those current owners who had sought information or advice, websites were the most common source of information (76%), followed by family and friends (70%), and online forums (51%). Similar patterns were seen among potential owners. Qualitative data revealed that information was commonly sought on the following: general information about dogs; information about breeds or types of dogs; owner requirements and dog suitability; aspects of dog ownership; and how to source a dog. The majority of prospective owners stated that they had found all the information they wanted (96% of current owners and 90% of potential owners) but some respondents found conflicting advice from different sources and did not know which sources to trust. Our study shows that, for some prospective owners, research into various aspects of dog acquisition and ownership is important. Knowing where to look for correct and unbiased advice, particularly online, is particularly important. Understanding the pre-acquisition research that owners undertake, including the sources they use, information they are interested in finding, and the challenges they face, is of interest to animal welfare and veterinary organisations and those involved in rehoming and selling dogs. This information may help inform interventions aimed at improving the knowledge and decisions made by potential dog owners.

## 1. Introduction

Dogs are the most popular companion animal species in the UK today [1], with many households aspiring to acquire dogs each year. Decisions made during the dog acquisition process can have widespread implications for canine welfare [2,3,4,5], but despite this, there is currently limited knowledge about how prospective dog owners decide how and where to find their dogs, as well as which dogs to acquire. Whilst large numbers of dogs are acquired every year, tens if not hundreds of thousands are relinquished to welfare organisations [6,7]. Reasons for this include owners’ lack of understanding of dogs’ needs and the time, effort, and costs involved in dog ownership [8,9,10,11]. For example, in a 2021 survey of UK dog owners, 52% reported that the cost of vet visits was higher than expected [12]. In the same study, many respondents described how they were surprised by the amount of resources, including time and money, that were required to fulfil a dog’s needs and meet the demands of dog ownership. It is possible that accessing information related to these aspects of dog ownership prior to acquiring a dog may increase prospective owners’ awareness of dogs’ needs, improve their expectations, and reduce the risk of relinquishment [11]. Various sources of information exist that provide advice about dog acquisition and ownership. These include advice from friends and family, breeders or vets, visiting events, e.g., [13], or reading books, magazines, online forums, social media posts, and webpages from individuals, groups, and companies. Indeed, in the UK, numerous charities and professional organisations have developed resources (predominantly web-based) that are designed to provide information for prospective owners about responsible dog acquisition and ownership, e.g., [14,15,16,17]. However, little is known about the extent to which resources such as these may influence subsequent acquisition behaviours and dog ownership [18]. This is an important area for many animal welfare organisations and professionals. Indeed, in a 2023 survey which asked veterinary professionals about their main welfare concerns, a third chose “lack of adequate pre-purchase education regarding suitable pet choice” as the issue they would choose to resolve tomorrow and nearly a quarter considered that this issue would have one of the biggest health and welfare implications in 10 years’ time if it was not tackled sooner [19]. Better understanding of pre-acquisition information-seeking is needed as part of efforts to inform the development of interventions aimed at improving decisions made by potential dog owners.

Previous research suggests that many prospective dog owners undertake some sort of research prior to acquiring a dog, e.g., [2,20,21]. A number of factors have been associated with this, including previous dog ownership experience (where those with more experience are less likely to look for information or advice) [20,22,23]. Where prospective owners do undertake research, we know little about what advice is sought, or what sources of information are considered. There is limited information on the quality of pre-acquisition research (i.e., the calibre of the research that prospective owners undertake prior to acquiring a dog). Neither is there much research into the time prospective owners spend on research, although a 2010–2011 survey found that 40% of puppy buyers spent one week or less conducting research prior to purchasing a dog [24].

Prospective dog owners may be influenced by various sources of information. A few studies have highlighted the influence of the internet in relation to pre-purchase behaviour with online resources being the most commonly reported source of information prior to acquiring their dog [25,26]. Taking advice from friends or family was also common, especially from those who own or had owned a dog [20,21,25,26]. Other popular sources included books and advice from breeders [21,26]. Little is known about the quality of information from these sources (i.e., the standard or correctness of the information provided).

Although not specific to pre-acquisition research, there have been a number of studies that investigated pet owners’ use of the internet to research pet health information, e.g., [27,28,29]. These highlight that although the internet clearly has benefits for pet owners, in terms of readily available resources on a wide range of topics, there are concerns about the quality of information [30]. Kuhl et al. [31] considered online information available about dog health that those who had recently acquired a dog may seek. They concluded that although there was an abundance of readily available information, most resources contained no supporting evidence, links to other resources, or reference lists. Although this does not mean that the information was inaccurate, the authors suggested the lack of reliability indicators may have hindered users’ ability to identify trustworthy sources to make welfare-compatible decisions. A study by Kogan et al. [28] found that over half of UK pet owners felt frustrated by a lack of information or an inability to find what they were looking for online, or they felt confused or overwhelmed by the information they did find. Despite this, 40% of owners viewed websites as trustworthy.

This study aimed to better understand the preparatory research undertaken by current and potential dog owners in the UK. Specifically, we were interested in understanding, prior to acquiring a dog: (1) where prospective owners look for information or advice; (2) what information or advice prospective owners search for; (3) how long prospective owners spend on research; and (4) whether prospective owners find the information they want.

## 2. Materials and Methods

This study collected data as described in Mead et al., 2023 [23], using a convergent mixed methods design. Data were collected through a survey and interviews as part of a wider study investigating various aspects of dog acquisition [31,32,33].

### 2.1. Ethics Statement

Ethical approval for this study was granted by the Dogs Trust Ethical Review Board (reference numbers: ERB018 and ERB019). Prior to participation, an informed consent statement was provided to all participants which outlined the purpose of the study, described how data would be stored and used, explained that participation was voluntary, and provided instructions on how to withdraw from the study. Participants were required to be aged 18 years or over and living in the UK. No payment or incentives were offered to participants.

### 2.2. Data Collection

#### 2.2.1. Survey Design and Content

An online survey was designed to collect data about the acquisition experiences of current and potential dog owners. The survey was hosted on the SmartSurvey^TM^ (https://www.smartsurvey.co.uk/ accessed on 4 October 2021) online platform and took approximately 20 min to complete. Questions were informed by a review of the current literature [34]. Survey items relevant to this study focused on pre-acquisition research experiences. Current owners were asked about any information or advice they had sought prior to getting their current dog and hence all information was retrospective. Potential owners were asked about research they had already undertaken and about plans for further research. To identify people who had undertaken research or, in the case of potential owners, planned to do so, respondents were asked whether or not they had looked for any information or asked anyone for advice before getting their dog. (The results of this study can be found in Mead et al., 2023 [23]). Current owners who selected “No”, and potential owners who answered “No and I don’t plan to” or “I haven’t thought about this”, were excluded from this study. Current owners and potential owners who selected “Yes” were asked questions about their experiences of pre-acquisition research, including what information or advice they looked for, where they looked for information or advice, how long they spent on research, and whether they were able to find all the information they wanted. Potential owners who selected “No but I plan to” were asked similar questions but these were worded to reflect the future nature of the potential owners’ plans. Respondents were required to answer all questions except where optional responses were specified. Data were also collected about the demographics of owners and (where applicable) their dogs. Additional methodological details can be found in Mead et al., 2023 [23], and all relevant survey questions and response options can be found in the Appendix A.

#### 2.2.2. Survey: Participant Recruitment

The survey was live between September and December 2019. Promotion was predominantly through Dogs Trust social media posts, correspondence with supporters, the Dogs Trust contact centre, and rehoming centres. The sample was therefore a convenience sample; however, some paid Facebook advertisement was used to target males and those who were not supporters of Dogs Trust. This was carried out to improve reach and to increase representation, given that male participants are under-represented in studies of human–animal relationships [35]. Further information on how participants were recruited to the survey can be found in the Appendix A.

#### 2.2.3. Interviews

Interviews were conducted with current and potential dog owners to gain a deeper understanding of aspects of the dog acquisition process. Each interview was conducted by one of three authors (R.M., K.E.H., or R.M.C). All interviews explored owners’ experiences of dog acquisition, including whether they had conducted any research prior to acquiring their current or prospective dog. Interviews followed a semi-structured guide which can be found in the Appendix A. With participants’ consent, interviews were audio recorded and later transcribed. In total, interviews were conducted with 24 current and 8 potential dog owners. Interviewees were recruited through the survey (*n* = 15), a pilot survey (*n* = 5), and pilot interviews with members of Dogs Trust staff (*n* = 12). Full details about data collection through interviews can be found in Mead et al., 2023 [23].

### 2.3. Data Analysis

#### 2.3.1. Quantitative Data Analysis

Initial data cleaning was completed in Microsoft Excel and IBM SPSS (v.26). Responses to relevant closed-ended survey questions were summarised with descriptive statistics using IBM SPSS (v.26) and R v.4.1.2; [36]. Chi-square tests were also completed using R (v.4.1.2). These were used to compare responses given by different groups of respondents.

#### 2.3.2. Qualitative Data Analysis

The purpose of this study’s qualitative analysis was to gain a deeper understanding of prospective owners’ experiences of conducting pre-acquisition research. Specifically, we wanted to identify the following: (1) additional information or advice sources consulted by prospective owners before getting a dog; (2) information topics sought by owners before getting a dog; (3) aspects of information that prospective owners struggled to find; and (4) additional understanding around how long owners spent on information and advice gathering. The aim was not to quantify these data.

Our approach to qualitative data collection and analysis aligned with a subtle realist perspective [37]. This position maintains that there exists a reality outside of, and knowable to, the researcher, whilst acknowledging that research is never independent of the perspective(s) of the researcher(s) involved.

Interview transcripts and relevant free text survey responses were imported into NVivo (v.12). Since interviews covered a variety of topics related to dog acquisition that were sometimes beyond the scope of this study, we first reviewed interview transcripts to identify material that was potentially relevant to this study. Relevant data were initially coded with the broad label “pre-acquisition research” ahead of being coded in greater detail to address this study’s research questions. The data from free text survey questions created a further fourteen data sets which were initially analysed separately. For instance, a similar question asking what information or advice did/would respondents look for before getting their dog was asked in three separate questions to different subsets of respondents, producing an initial data set for each subset: (1) current owners who reported having completed research; (2) potential owners who reported having already completed research; and (3) potential owners who reported planning to undertake research. Later, data sets for similar questions were combined and refined as a whole.

Data were analysed using a codebook approach to thematic analysis, similar to the template analysis approach [38]. This structured form of thematic analysis was in keeping with this study’s pre-determined information needs and is well suited to team-based working. Different portions of the data were coded by two data analysts (R.M. and K.E.H.) who both had prior personal experiences of acquiring dogs in the UK and work for the UK’s largest dog welfare charity.

We followed procedural steps similar to those outlined by King [38]. Whilst it is common in studies using a template analysis style of thematic analysis to begin the coding process with some a priori themes, considered to be useful or relevant for the analysis, we did not start our initial coding in this way. Instead, we maintained a more open attitude to developing codes and themes, generating them inductively. As coding progressed, initial codes were grouped into meaningful categories that consolidated data and then organised hierarchically, with narrower themes capturing greater detail subsumed within broader ones (e.g., “behaviour” and “size” were subsumed under the main theme “dog characteristics”, which was itself encompassed by the overarching theme “information about dogs”). As later data items were coded, where necessary, the initial codebook was modified.

Where subsets of responses to individual survey questions were coded by both analysts, the final themes and their different levels (i.e., “overarching-”, “main-” and “sub-themes”) were established collaboratively through reviews and discussions of the respective coding by both authors, sometimes in conversation with a third author (R.M.C.). These discussions helped to deepen our engagement with the data, for instance by highlighting any overlooked areas within our respective analyses. Working collaboratively also encouraged reflection on justification for the inclusion of each theme, consideration of alternative interpretations, and some modifications throughout the process, for instance broadening or narrowing a theme. The final codes and themes for each data set constituted our codebooks, which were used to describe the data.

A large quantity of data was collected across the 14 free text survey questions. Four questions received >1000 responses each. Owing to pragmatic considerations (i.e., time constraints), analysts considered the “information power” [39] of the responses to each question during the coding of the first 1000 responses. For all four questions, the researchers determined that a subsample of responses was sufficient to meet the aim of this study’s qualitative analysis. A quasi-random sampling approach was then applied to the remaining data, which involved coding every 25th additional response. Whilst we remained open to adding new codes or refining existing ones throughout the coding process, when coding the additional (every 25th) responses, it was very uncommon for new codes to be created. This is not to say that coding of further data would not have led to further refinements to coding, but it indicates that the developed codebook was sufficient [40]. Further details on which qualitative survey data were coded, and how this was split between the analysts, are presented in the Appendix A.

## 3. Results

### 3.1. Survey Results

This study reports on a subset of 4382 current owners and 2350 potential owners who had undertaken, or planned to undertake, research prior to acquiring a dog. This subset represented 54.4% of current owners and 81.5% of potential owners (67.8% had already undertaken research and 13.7% planned to) who completed the survey (these findings, along with factors that affected the likelihood of undertaking research, using data from this project, are reported in Mead et al., 2023 [23]).

#### 3.1.1. Participant Demographics

The majority of survey participants were female (88.9% of current and 80.1% of potential owners). Respondents represented age groupings from 18 to 85 years and above, with 45–54 years being the most common age category for current owners (23.7%) and 25–34 years being the most common for potential owners (21.6%). Respondents resided in all four UK nations, but the majority were based in England (83.4% of current and 82.8% of potential owners where a postcode was provided). Additional information on participant demographics can be found in the Appendix A.

#### 3.1.2. Dog Demographics

Two thirds of current owners (66.1%) had acquired their dog within 5 years prior to the completion of the survey. Over half of dogs (61.0%) were acquired as puppies of 6 months or younger. Over half (57.6%) were a specific breed (e.g., Labrador Retriever) and 23.1% were a mix of two specific breeds (e.g., CockerpooT/Cocker Spaniel × Poodle cross). Most dogs were acquired from a dog breeder (48.5%) or from a charity or rehoming centre (37.8%). Further information on dog demographics can be found in the Appendix A.

### 3.2. Where Do Prospective Owners Look for Information?

Most respondents reported using multiple sources when undertaking research. Of the 15 possible options provided, the median number of sources selected by current owners was four (min = 1, Q1 = 2, Q3 = 5, max = 12). This was similar for potential owners who had conducted research (min = 1, Q1 = 2, median = 3, Q3 = 5, max = 13) and those who planned to undertake research (min = 1, Q1 = 2, median = 4, Q3 = 5, max = 15). Websites were the most common source of information among current owners with three quarters citing their use (76.2%; Table 1). Other popular sources were family or friends (69.6%), online forums (51.1%), and books (37.3%). Although there were similar patterns between the three ownership categories, significant differences occurred between these groups for nearly all sources. Potential owners were more likely to have spoken to family and friends than current owners (84.8% vs. 69.6%). Potential owners who had not yet undertaken research had higher aspirations of using canine professionals than seen among potential and current owners who had undertaken research. This included vets (29.2% of potential owners planned to speak to vets compared with 12.5% of potential and 13.8% of current owners who had sought information from vets) and behaviourists or trainers (22.0% vs. 11.5% and 12.3%, respectively). The sources of information consulted by current owners also appeared to differ by age category, gender, and acquisition source (Appendix A).

Data from free text responses and interviews offered further insight into why prospective owners used certain sources. The importance of family and friends was clear for many prospective owners. Responses indicated that advice was often sought from these people because of their experiences of dog ownership. For example, in this quote, the interviewee was planning to get a dog from a rescue centre:


*“I spoke to quite a lot of my friends who’ve got dogs. I’ve got friends who have and volunteer with rescue dogs. I’ve got quite a few friends that have got dogs of every breed to be honest. We did a lot of speaking to different people and talking about their dogs’ personalities.”*
(Potential owner, interview ID B2RM0301)

In some cases, prospective owners appeared to rely completely on family members to help with decision making, especially where they were professionals in the field or considered experienced. As in the example below, this sometimes enabled advice or information to be personalised, where the person consulted knew the prospective owner well, including in the following example:


*“I consulted with my daughter who is a highly qualified veterinary nurse […] She knows me best and knew what sort of dog would be good for me.”*
(Current owner, survey ID 1770)

Some prospective owners appeared to have a desire for others’ first-hand experiences of owning a particular breed or of using a specific source. This included using online sources, such as blogs, forums, and social media, as well as speaking to friends, family, or other dog owners:


*“Spoke to people who owned the breed.”*
(Current owner, survey ID 1929)


*“Talked to people with experience with rescues in the UK.”*
(Current owner, survey ID 1991)

A number of prospective owners discussed more immersive research, such as looking after a friend’s dog to gain first-hand experience of what life with a dog might be like, including the following:


*“A colleague has two Cocker Spaniels… I had one for a taster day in my own home.”*
(Current owner, survey ID 7586)

Hence, prospective owners placed considerable value on personal knowledge and lived experience, and our findings suggest that personal ownership experience may be prioritised over more formal or impersonal sources of knowledge for many prospective owners.

A wide range of specific online resources was identified, such as websites of the following: breed-specific groups; charities and rehoming organisations, including breed-specific rescues; general dog information websites; and pet selling sites. Some specific books were named. These often related to specific breeds or a particular life stage, with resources focused on puppies appearing frequently.

### 3.3. What Information Do Prospective Owners Report Searching for?

Analysis of the qualitative data from the free text responses and interviews revealed that prospective owners sought (or planned to seek) information encompassed by five overarching themes: (1) information about dogs in general; (2) information about breeds or types of dogs; (3) owner requirements and dog suitability; (4) aspects of dog ownership, and (5) how to source a dog. These are presented alongside illustrative quotations in the Appendix A, and are discussed below.

#### 3.3.1. Information about Dogs

Information was sought within two general categories: the needs of dogs and characteristics of dogs. Prospective owners who were interested in the needs of dogs typically sought information related to food and diet, exercise or activity levels, grooming, and health. Searches for information about the amount of space needed for a dog, the size of a dog, and dogs’ needs in general were also common. Usually, prospective owners sought information related to multiple aspects of dogs’ needs:


*“What diet the dog needs, ideal exercise routine, how much space a dog will need in the house and the garden.”*
(Potential owner, survey ID 3060)

Dog characteristics focused on understanding what personality, temperament, or traits a dog might have. It was not always clear whether these referred to a specific dog or dogs in general. Occasionally, prospective owners referred to the behaviour, size, or trainability of dogs. Often, prospective owners wanted information about many aspects of dogs’ needs and characteristics, such as this survey respondent:


*“Information on size, temperament, health issues, exercise requirement, grooming requirements, difficulty of training.”*
(Current owner, survey ID 2775)

In the data encompassed within this theme, owners sometimes referred to specific life stages, such as puppies or older dogs, and occasionally, participants were interested in information specific to rescue dogs.

#### 3.3.2. Information about Breeds or Types of Dogs

Many prospective owners were interested in learning more about specific breeds or types of dogs. Information was categorised by two themes: choosing a breed or type of dog and information about a particular breed or type of dog. These two themes are illustrated in the quote below, where initially, this owner looked broadly at different breeds that they might have been interested in owning before narrowing this down to two breeds and looking for information specific to these:


*“We looked at breeds that we liked and weighed up pro’s and con’s [sic.] for all, and narrowed it down to two breeds, and then did extensive research on both.”*
(Current owner, survey ID 1910)

Where prospective owners wanted general information about different breeds or types, this was sometimes linked to factors such as owners’ lifestyles, with owners wanting to understand what breeds might suit them best or be the most compatible for their lifestyle. Some prospective owners already had ideas about breeds they were interested in and wanted to learn more or narrow their options down. Others knew there was one specific breed they were interested in and wanted information on that breed. Sometimes, this was linked to a specific dog, for example, in this quote, the interviewee had recently acquired a Staffordshire Bull Terrier from a rehoming centre:


*“We did do quite a lot of research about Staffies and what not because none of us have actually ever had Staffs [sic.]. So, we just wanted to know a little bit about what habits they have. I looked into a lot of things like the best way to train them, certain behaviours, what sort of diets were best for them.”*
(Current owner, interview ID B1RM1104)

Key areas that were commonly mentioned related to wanting information about a particular breed or type of dog included the following: breed characteristics, temperament, or traits; health and, in particular, health concerns; how much a breed’s coat might shed and/or whether a breed had a “hypoallergenic” coat type; and what training a particular breed might require or how trainable that breed might be. In addition, the needs of the dog were also commonly mentioned but in terms of the feeding, exercise, and grooming requirements of a particular breed. Many prospective owners wanted to know about lots of different aspects specific to a breed, as can be seen in this quote:


*“Online researching the breed. Health defects, temperament, how much food they eat, common genetic issues, amount of exercise and space required, how much they shed.”*
(Current owner, survey ID 241)

Occasionally, prospective owners considered the characteristics or needs of breeds in terms of what they could *not* manage, such as this survey respondent, who wanted to ensure they could meet the exercise needs of a potential dog:


*“If there were any breeds which needed more activity than I could provide.”*
(Current owner, survey ID 1757)

In rare cases, prospective owners reported changing their mind about a particular dog or breed based on their research or, in the following example, following a home visit from a dog through a rehoming centre. For these prospective owners, this seemed due to their concerns for the dog’s wellbeing and their ability to meet his/her needs:


*“We had a home visit from a Staffy [Staffordshire Bull Terrier], which made us realise that is [sic.] wasn’t the breed for us. Because we learnt that Staffies love attention and kept active due to the breeds intelligence. And because of our life style we understand that this breed would be unhappy living with us.”*
(Current owner, survey ID 1208)

#### 3.3.3. Owner Requirements and Dog Suitability

Many prospective owners discussed their requirements from a dog. Terms related to “*suitability*” were often used, with respondents wanting to find out about an individual dog and/or breed or type to assess how *suitable* they may have been to their circumstances. Family, especially babies and children, were important considerations for many prospective owners. Some reported needing a good “*family dog*” or wanting a “*calmer*” dog breed:


*“I looked at breeds which were good around young children. My previous dog was a Jack Russell and whilst I wanted another one we decided on a spaniel due to their calmer nature.”*
(Current owner, survey ID 1080)

A number of prospective owners discussed their living arrangements, with some noting that they required a dog of a certain size to fit within their house or flat. Mention of a living space and garden (or lack thereof) were particularly prominent among potential owners, where their research often appeared to question whether their home setup was suitable for a dog, with some potential owners wanting to understand how they may have needed to adjust their home and/or garden to ensure it would be suitable for a dog:


*“The suitability of our house and garden for a dog, whether we needed to change anything there.”*
(Potential owner, survey ID 1325)

Many prospective owners discussed their lifestyle and how they needed a dog that would be compatible with it. This included consideration of a dog that would meet an owner’s (and, where relevant, their family’s) activity levels, how long a dog could be left alone while owners were out, typically at work, and learning more about taking a dog to work. Particularly among potential owners, research focused on understanding how to manage work commitments whilst owning a dog, with some even raising the question of whether they should own a dog at all, given that they worked full-time:


*“Whether it’s appropriate when you work full time.”*
(Potential owner, survey ID 1627)

Other animals, including dogs and cats, were often mentioned. Some who already owned dogs wanted advice about whether they should acquire an additional dog. They were often interested in information about how owning two dogs might be different to one, how to introduce a new dog to their existing dog(s), and how best to ensure a new dog would be a good match for an existing dog or dogs. The suitability of a dog or breed was also mentioned with respect to other animals, including chickens and horses, but mainly cats. Some prospective owners wanted to know if there were particular breeds or types of dogs that were more likely to be compatible with cats. Others were interested in how to introduce a dog to a cat.

Some prospective owners used a combination of their requirements as a guide to finding the best type of dog for their circumstances:


*“I would need a dog who is cat and child(older) friendly. I look at websites to try to establish the best age and temperament for me and my family.”*
(Potential owner, survey ID 1172)

As can be seen in the above examples, generally, research was human-focused in terms of what the prospective owner and sometimes their family (including other animals) needed from a new dog. However, some prospective owners were more focused on whether they, and their circumstances, were suitable for a dog. Some considered the suitability or “*match*” from both human and dog perspectives:


*“How to find a perfect match for the dog with the home and circumstances I have to offer. The match is my overall priority and concern.”*
(Potential owner, survey ID 1223)

#### 3.3.4. Aspects of Dog Ownership

Many prospective owners were interested in knowing more about the experience and practical aspects of dog ownership. This often seemed to be driven by a desire to be prepared for the arrival of a dog and for dog ownership in general. Some people sought to understand what it would really be like to own a dog and what ownership might involve, again highlighting the importance of lived experience and personal knowledge for many. This included consideration of the time commitment and responsibilities dog ownership would bring about. Some were interested in understanding how having a dog might impact their lifestyle or routine. A small number of respondents were keen to learn about the more challenging aspects of dog ownership and not just the positive ones:


*“How friends and family have found their lifestyle altered by having a dog, both in positive and negative ways.”*
(Potential owner, survey ID 1158)

Aspects of caring for a dog throughout their life, such as vet care, vaccinations, and insurance, were important considerations for many prospective owners. These were often related to the cost of owning a dog. Support networks were discussed and included dog walkers or day care and friends or family who would help care for a dog. There was an emphasis on seeking local services or facilities:


*“What support I could find locally (boarding, help with exercise, etc as we have limited family support nearby).”*
(Current owner, survey ID 3259)

Dog training was mentioned by many. Sometimes, this referred to learning about training through books, videos, and online resources, but consideration of local training classes was also common. Learning more about care relevant to specific life stages or experiences (such as rescues) or specific activities such as agility, and the local provision of these, were also discussed by some:


*“Information about puppy socialisation, training classes and positive reinforcement training techniques and programmes, further enrichment training for adult dogs like agility and scent classes.”*
(Potential owner, survey ID 1798)

#### 3.3.5. How to Source a Dog

Some respondents researched how and where to get a dog, typically from a breeder or a rehoming centre. Prospective owners who were interested in sourcing a dog from a breeder commonly wanted to know how to find reputable, responsible breeders and avoid puppy farms. Some mentioned seeking breeders who were registered, accredited, or assured. Usually, this was in reference to the UK Kennel Club Assured Breeders scheme [41]. Others wanted to know what to look for when visiting breeders or what questions to ask breeders.

Other respondents were interested in researching rehoming centres and the process of adopting a dog, including what policies or requirements rescue organisations may have, and whether they would be considered suitable adopters. Looking at available dogs was considered research by some prospective owners. Some were interested in learning about rehoming from outside of the UK. This included researching overseas rescue organisations to ensure they were reputable and understanding whether overseas rescue dogs might need special support or “*back-up*”, as in this example:


*“I spoke to many rescue centres (a lot of smaller ones) in the search to find a suitable dog. It took 1.5 years to find the right rescue dog as nothing was suitable from UK rescue […] then we found a puppy from an overseas rescue which has been a perfect fit. We then did a lot of research into the overseas charity and wanted to make sure it was a responsible one with back-up should anything go wrong. I also asked advice from friends and clients who have adopted dogs from overseas.”*
(Current owner, survey ID 7359)

As seen in the examples above, our five overarching themes tended to be interlinked, particularly around *suitability* or *compatibility*. Many respondents were interested in multiple aspects of research prior to acquiring a dog, and some spoke about wanting information about “*everything*”. The variety of information—and sources—some prospective owners sought is highlighted in the following quote:


*“Everything I could! Researching how to care for and train a puppy. Talking to friends with well cared for and well trained dogs. Researched health needs and possible health problems. Researched dog training. Researched different breeds and types of dog to find a good match. Talked to local [dog rehoming centre] (and looked their [sic.] for a suitable dog before then looking for a GOOD breeder).”*
(Current owner, survey ID 5082)

### 3.4. How Long Do Prospective Owners Spend on Research?

For many prospective owners, researching aspects of dog acquisition and ownership appeared to take a considerable amount of time: of those who sought information or advice, over 40% of current and potential owners conducted research over two to six months. Although there were similar patterns across the groups, potential owners who had already undertaken research tended to report having spent longer doing this: nearly a quarter had looked at information for over a year or more compared to less than 2% of potential owners who were yet to conduct any research (Table 2).

Although these times give an idea of the length of time or the period over which prospective owners considered factors related to dog acquisition, it was difficult to tell what the length of time actually meant, as noted by one respondent:


*“It is difficult to put a time on this, it is dependent on the quality of the information from each source.”*
(Potential owner, survey ID 4477)

Some potential owners described dipping in and out over a period of months or even years and this appeared to be more part of a contemplative or preparatory stage:


*“I have been considering looking for about 6 months but wanted to make sure I was 100% ready before I began to look which I now know I am.”*
(Potential owner, survey ID P4044)

Some respondents gave more details, such as this respondent, who noted that they conducted small amounts of research over a number of years while waiting for the “*right time*” or for the right circumstances:


*“Lots of small amounts of time over several years, focussed research when feeling the time might be right to make sure I’d considered everything.”*
(Current owner, survey ID 1414)

For some, dog ownership appeared something of a life ambition which they had been planning for, and gathering information about, over many years:


*“I’ve researched as much as I can from the age of five, I’ve dreamt about it for 23 years so there was alot [sic.] of time to research!”*
(Current owner, survey ID 8136)

Some mentioned their general interest in dogs and keeping up-to-date with information, while others noted that the time spent on research differed as they became more experienced dog owners. For others, research only appeared to begin once they knew what dog or breed they were getting, including the following example:


*“Once we knew we were getting [dog’s name]. We did lots of research into dog care in general and further research into his breed. We wanted to ensure we had a good knowledge and understanding of everything he would need in relation to health and well-being specifically. Making sure we knew how to train a dog properly and we knew how to read their body language and assess situations to ensure safety. We were very thorough in finding our [sic.] everything we could about being a dog owner.”*
(Current owner, survey ID 1696)

Some mentioned a change of pace when they knew they were in a position to get a dog, with some noting that the circumstances around how the dog was acquired dictated the time that could be spent on research. This seemed particularly important for those who rescued a dog in need and had limited time to act, including the following example:


*“Sounds irresponsible but in rescue situations there is not the luxury of long searches for information.”*
(Current owner, survey ID 9173)

### 3.5. Do Prospective Owners Find the Information They Want?

The majority of prospective owners found the information they were looking for with 90.2% of potential owners and, retrospectively, 95.7% of current owners stated that they had found all the information they were seeking.

Qualitative responses revealed some insights into what information was *not* found. Of those who had not found the information they wanted, the areas that advice was needed in were varied. Queries around adopting a dog and family or work circumstances seemed reasonably common, such as whether prospective owners with young children or who worked full-time would be able to adopt a dog.

However, sometimes these involved quite specific questions or situations, such as how to train dogs with specific needs, such as deaf dogs.

Some prospective owners commented on how they had found conflicting advice from different sources and that they did not always know which sources or information to trust. Some reported having to look in lots of different places to find all the information they needed, such as this prospective owner:


*“The information doesn’t seem to be complete and comprehensive resulting in research from lots of different locations and piecing it together.”*
(Potential owners, survey ID 8245)

Typically, these seemed to refer to online sources, including, for example:


*“I wasn’t sure about the quality of advice I was reading. Many websites contradicted other sites.”*
(Current owner, survey ID 2367)

## 4. Discussion

This study used mixed methods to understand where prospective owners looked for information or advice, and what information was sought, prior to acquiring a dog. Similar to previous research, e.g., [20,21,25,26], our study found that prospective owners reported being most likely to find information online or from friends or family. As has been noted by other studies, e.g., [28,29,30,31], although the internet has many benefits, such as being readily available to many and covering a vast array of topics, there are concerns about the reliability and completeness of information it provides. Owner concerns about identifying accurate and trustworthy information were highlighted in our qualitative research. The importance of friends or family who had the experience of dog ownership, who had owned a particular breed or type, and particularly those who were considered dog professionals was clear. Whilst many such people may provide valuable information, there will be variation in the extent to which this advice is evidence-based or in keeping with current best practice. Despite this apparent trust in friends and family who were considered to have relevant expertise or experience, few prospective owners consulted professionals such as vets or behaviourists. We found that less than 14% of prospective owners had consulted a vet. Smaller proportions have been reported in previous studies, including a 2017 survey that found only 6% of owners reported taking advice from veterinary professionals prior to acquiring a dog [26] and a 2020 survey that found just 3% of owners had undertaken a free online consultation with a vet prior to getting a pet [25]. Speculatively, Kuhl et al. [21] suggest that owners may not feel comfortable consulting with vets before acquiring an animal, may not be aware that veterinary professionals offer this support, or may be concerned about costs for such services. Prospective owners may also be driven by convenience consumer behaviour, leading to preference for seeking information that may be immediately accessible online. Despite this, there is clearly some interest in pre-purchase support from veterinary practices: within the same 2020 study, 45% of surveyed owners indicated that they would be interested in a free online consultation with a vet before acquiring a pet [25]. Indeed, nearly 30% of potential owners in our study reported that they were planning to seek advice from a vet prior to acquiring a dog. Pre-purchase consultations are championed by welfare organisations such as the UK veterinary charity, the People’s Dispensary for Sick Animals (PDSA) [42]. However, in the 2018 PDSA Animal Welfare (PAW) report, data collected from the British Veterinary Association (BVA) and British Veterinary Nursing Association (BVNA) reported that only 13% of veterinary practices offered free, dedicated, pre-purchase clinics or appointments [43]. Lack of time, limited staffing, the need for practices to be profitable, and perceived public distrust in the veterinary profession have been identified as barriers to veterinary clinics engaging in pre-purchase consultations regarding brachycephalic dogs, but clearly more general research is needed into this area [44]. This might suggest that more could be done to promote professionals, such as vets, as sources of quality information. Whilst it is encouraging that most prospective owners use a range of sources when undertaking research, we do not know how much value is placed on different sources. There were differences between the sources used by those who had undertaken research and the aspirations of those who planned to, particularly in relation to consulting animal professionals such as vets, dog behaviourists, and trainers. This may suggest that this aspiration is often not acted upon prior to obtaining a dog. More research is needed to better understanding the barriers and drivers that affect engagement with professionals, prior to the acquisition of a dog [18].

Our study provides insights into the types of information that prospective owners seek. The importance of dog breed is clear, with owners placing considerable significance on perceived distinctions between different breeds and types of dogs. For some prospective owners, their research involved trying to choose or narrow down a breed or type that might be suitable for them. For others, they had already chosen a particular breed and wanted more information on that breed. Choosing a dog based on their breed or type may be sensible when considering practicalities such as their space and exercise requirements, but basing this choice on “character”, “temperament”, and “traits”, which were terms frequently used by prospective owners, is more concerning. There is a growing body of research suggesting that many behavioural characteristics vary more between individual dogs within a breed than between breeds, e.g., [45]. Although some prospective owners stated that they researched differences between breeds and made acquisition decisions based on these, this information could have been misleading. For example, choosing a breed that was “good with children” could lead to unrealistic expectations and insufficient consideration of the need for appropriate training and active supervision for all dogs around young children. Belief in breed stereotypes appeared common among our prospective owners and might have detracted interest from what could have been potentially suitable individual dogs. Such stereotypes may lead to more fashionable—but potentially less healthy—breeds receiving considerable attention from prospective owners, whereas temperamentally suitable individual dogs of less popular or mixed breeds or types receive less interest. This could be particularly important in a rehoming setting.

Another important theme is the suitability of dogs to owner’s circumstances. Our qualitative data revealed that owners place great importance on finding “suitable” dogs and that they often require and expect a lot from their dogs. Although it is encouraging that many prospective owners seem to consider what they need from a dog and how best to find a dog that matches their requirements, to some extent, this places a lot of pressure on any future dog to fulfil these expectations. Where these expectations are not met, dogs may be at risk of relinquishment or euthanasia [9,46,47]. Fewer people considered whether what they offered would be suitable for a dog. Although some mentioned working hours and dogs being alone for long periods, very few specifically discussed how this could be managed, such as with the assistance of dog walkers or sitters. Most seemed to acknowledge this was not ideal for a dog; however, they were still eager to proceed with acquiring a dog.

Although owners seek information on a variety of topics, understanding whether they are looking for and find the “right” information is complicated. The majority of prospective owners who undertook research reported finding all the information they wanted. However, it is difficult to know how to interpret this for the following reasons: we do not know how accurate the information they found was nor did they necessarily know what they needed to know insofar as they may have been unaware of topics which could have benefitted them and their future dogs. In our study, considerably more respondents appeared satisfied with the information or advice they had found than did those surveyed by Kogan et al. [28] Their study found that over 50% of pet owners felt “frustrated by a lack of information or an inability to find what I was looking for”, “sometimes”, “often”, or “almost every time”. Although the two studies are not directly comparable, as Kogan et al.’s study investigated the use of the internet only to source health information for current pet (not just dog) owners and questions asked were different, these contrasting findings suggest an area worthy of further exploration.

Attempting to evaluate the quality of the research that prospective owners undertake is challenging and not an aim of this study (but this would be an interesting area for future research). What constitutes research to one person may be something very different to someone else. Although the time spent looking for information before acquisition reported here gives an idea of how long prospective owners might spend considering factors related to dog acquisition, it is difficult to tell what the length of time actually means. For example, does “six months” mean a prospective owner spent many hours every week over a period of six months looking for information, or did that person very occasionally ask for advice over half a year? Considering length of time spent on research may not be particularly meaningful, but it does give an indication of the timeframes over which people may be open to receiving information. It does not reflect the quality of research, nor the amount of time spent on specific resources. Regardless, our study found that considerably more time was reported to be spent on pre-acquisition research than previous studies in this area. A study by the RSPCA found that 40% of owners spent one week or less conducting pre-acquisition [24]. In comparison, 87% of current owners and 93% of potential owners in our study recalled spending longer than a week engaged in pre-acquisition research.

Although research often seems to fit within a pre-acquisition stage, and acquisition behaviours are often informed by this, it is important to recognise that research is something that could occur at different and even multiple timepoints. For example, some participants mentioned particular timepoints at which they began their research, such as when they knew which breed of dog they were acquiring. Additionally, research is likely to continue into dog ownership, after dog acquisition [21]. Better understanding of this pre-acquisition research may be useful for those hoping to influence canine welfare prior to dog acquisition and across a dog’s life.

This study provides novel insights into the information that prospective owners seek and also offers insights into how this information was sought and the timescales over which this research was undertaken. Our findings have implications for organisations and professionals involved in canine welfare and the provision of pre-acquisition advice. These stakeholders may want to think about the following: whether their advice includes the topics highlighted as of interest within this study; the quality of any resources; and the means by which they provide information. Dog professionals, such as vets, may wish to consider if and how they could better market themselves as offering pre-acquisition consultations to prospective owners.

### Strengths and Limitations

This study was the UK’s largest published study of pre-acquisition research among prospective dog owners of which we are aware. Using a mixed methods approach allowed quantitative and qualitative data to be integrated. The inclusion of free text survey items allowed for a wider coverage of responses, whereas semi-structured interviews provided deeper insights into pre-acquisition research behaviours. Collecting data from both current and potential owners enabled retrospective and current perspectives to be observed.

However, this study has several limitations. The methods involved using a convenience sample, with a bias towards Dogs Trust supporters. Respondents were self-selecting and there was a predominance of female respondents. This is common in studies related to human–animal relations [35], but this underrepresentation of males and those who prefer to self-identify needs further research. All survey responses were self-reported; thus, it was not possible to validate findings. There was the potential for recall bias, particularly among current owners who had owned their dog for some time. Although most closed survey questions included “other” free text boxes, it is possible that providing more response options would have elicited additional responses. For example, rescue centre staff or adoption advisors were not included as response options in the question related to where prospective owners sought information or advice. It is possible these sources were used but not recorded as they were not offered as answers in the survey. Future research may like to consider additional options such as these. Although a large amount of data was used for qualitative analysis, only a relatively small proportion of this was obtained through interviews. Additional interviews might provide a deeper understanding of pre-acquisition behaviours. A cautionary note should be applied to the analysis of data related to potential owners who planned to undertake research: whilst these data can be used for hypothesis generating, further research through follow-up surveys will be used to understand whether they completed the planned research.

## 5. Conclusions

Undertaking research before acquiring a dog is important for, and valued by, many prospective owners. Online resources are commonly used and the advice of family and friends is also valued by many, particularly where these are dog professionals or are considered to be experienced dog owners. However, dog professionals such as vets, are less commonly consulted with regards to pre-acquisition advice. Prospective owners want information related to many areas of dog care and ownership. These include the requirements of dogs and finding out about breeds and types of dogs, in terms of their requirements but also breed traits and health issues. These are often linked to an owner’s requirements, such as suitability to their lifestyle. Issues related to dog ownership are also important and include a consideration of costs. Finally, some prospective owners want help in understanding how to source a dog responsibly. Most owners reported finding all the information that they wanted. However, some raised concerns about the quality of information they found and conflicting advice. They were not always sure which sources to trust. Knowing where to look for correct and unbiased advice, particularly online, is of key importance. Our findings may be of interest to organisations involved in canine welfare, especially those who provide advice related to dog acquisition and dog ownership, in order to improve the knowledge and decisions made by prospective owners.

## Figures and Tables

**Table 1 animals-14-01033-t001:** Sources of information for those respondents who were current owners and responded retrospectively (*n* = 4381), for those who were potential owners and had already undertaken some research (*n* = 1955), and for those potential owners who were planning to look for information (*n* = 395). Respondents were able to select multiple responses. Results with differing superscript letters are statistically different (*p* < 0.05).

Source of Information or Advice	Current Owners			Potential Owners (Used)			Potential Owners (Plan to Use)			Statistics			
	*n*	%	95% CI	*n*	%	95% CI	*n*	%	95% CI	*X* ^2^	*p*-Value	*DF*	*Effect Size*
Family or friends	3051	69.64% ^a^	68.26%, 70.99%	1657	84.76% ^b^	83.09%, 86.28%	282	71.39% ^a^	66.74%, 75.63%	162.71	<0.001	2	0.16
Websites	3340	76.24% ^a^	74.96%, 77.48%	1345	68.80% ^b^	66.71%, 70.81%	293	74.18% ^a^	69.64%, 78.25%	38.86	<0.001	2	0.08
Online forums	2240	51.13% ^a^	49.65%, 52.61%	763	39.03% ^b^	36.89%, 41.21%	195	49.37% ^a^	44.47%, 54.28%	79.96	<0.001	2	0.11
Books	1633	37.27% ^a^	35.85%, 38.72%	344	17.60% ^b^	15.97%, 19.35%	125	31.65% ^c^	27.25%, 36.39%	243.78	<0.001	2	0.19
Social media	114	26.09% ^a^	24.81%, 27.41%	449	22.97% ^b^	21.16%, 24.88%	102	25.82% ^a,b^	21.75%, 30.36%	7.10	0.029	2	0.03
TV	562	12.83% ^a^	11.87%, 13.85%	446	22.81% ^b^	21.01%, 24.73%	42	10.63% ^a^	7.94%, 14.08%	110.23	<0.001	2	0.13
Vet	603	13.76% ^a^	12.78%, 14.82%	245	12.53% ^a^	11.14%, 14.08%	115	29.11% ^b^	24.85%, 33.78%	76.72	<0.001	2	0.11
Blogs	628	14.33% ^a^	13.33%, 15.4%	233	11.92% ^b^	10.55%, 13.43%	66	16.71% ^a^	13.34%, 20.72%	9.69	0.008	2	0.04
Dog breeder	727	16.59% ^a^	15.52%, 17.73%	106	5.42% ^b^	4.50%, 6.52%	35	8.86% ^c^	6.41%, 12.1%	156.30	<0.001	2	0.15
Dog behaviourist/training professional	538	12.28% ^a^	11.34%, 13.29%	224	11.46% ^a^	10.12%, 12.95%	87	22.03% ^b^	18.21%, 26.38%	34.56	<0.001	2	0.07
Events	447	10.20% ^a^	9.34%, 11.14%	185	9.46% ^a^	8.24%, 10.84%	77	19.49% ^b^	15.88%, 23.7%	36.54	<0.001	2	0.07
Member of public	355	8.10% ^a^	7.33%, 8.95%	220	11.25% ^b^	9.93%, 12.73%	24	6.08% ^a^	4.08%, 8.92%	20.67	<0.001	2	0.06
Magazines	397	9.06% ^a^	8.25%, 9.95%	93	4.76% ^b^	3.89%, 5.80%	25	6.33% ^a,b^	4.29%, 9.21%	36.49	<0.001	2	0.07
Dog walker/sitter	247	5.64% ^a^	4.99%, 6.36%	174	8.90% ^b^	7.71%, 10.25%	41	10.38% ^b^	7.72%, 13.80%	30.62	<0.001	2	0.07
Dog groomer	109	2.49% ^a^	2.06%, 2.99%	45	2.30% ^a^	1.72%, 3.07%	12	3.04% ^a^	1.69%, 5.29%	0.77	0.682	2	0.01
None of the above	160	3.65% ^a^	3.13%, 4.25%	68	3.48% ^a^	2.75%, 4.39%	NA	NA	NA	NA	NA	NA	NA
Haven’t thought about	NA	NA	NA	NA	NA	NA	8	2.03%	0.96%, 4.02%	NA	NA	NA	NA

**Table 2 animals-14-01033-t002:** The approximate times taken by prospective owners to find information or advice, prior to acquiring a dog, current *n* = 4381, potential have undertaken research *n* = 1955, potential plan to undertake research *n* = 395. Results with differing superscript letters are statistically different (*p* < 0.05).

Time Spent on Research	Current			Potential (Undertaken Research)			Potential (Plan to Undertake Research)			Statistics			
	*n*	%	95% CI	*n*	%	95% CI	*n*	%	95% CI	*X* ^2^	*p*-Value	*DF*	*Effect Size*
Up to a day	253	5.77% ^a^	5.12%, 6.51%	32	1.64% ^b^	1.15%, 2.31%	39	9.87% ^c^	7.28%, 13.24%	73.97	<0.001	2	0.10
More than a day, up to a week	271	6.19% ^a^	5.51%, 6.94%	48	2.46% ^b^	1.85%, 3.25%	27	6.84% ^a^	4.71%, 9.80%	41.05	<0.001	2	0.08
More than a week, up to a month	1329	30.34% ^a^	28.99%, 31.71%	362	18.52% ^b^	16.86%, 20.30%	131	33.16% ^a^	28.70%, 37.95%	103.55	<0.001	2	0.12
More than a month, up to six months	1873	42.75% ^a^	41.29%, 44.22%	790	40.41% ^a^	38.25%, 42.60%	160	40.51% ^a^	35.78%, 45.42%	3.40	0.1824	2	0.02
More than six months, up to a year	233	5.32% ^a^	4.69%, 6.02%	192	9.82% ^b^	8.58%, 11.22%	13	3.29% ^a^	1.88%, 5.60%	52.18	<0.001	2	0.09
Longer than a year	357	8.15% ^a^	7.37%, 9.00%	467	23.89% ^b^	22.05%, 25.83%	7	1.77% ^c^	0.79%, 3.69%	352.77	<0.001	2	0.23
Not sure or other	65	1.48% ^a^	1.16%, 1.89%	64	3.27% ^b^	2.57%, 4.16%	18	4.56% ^b^	2.86%, 7.13%	31.34	<0.001	2	0.07

## Data Availability

The data presented in this study are available upon request from the corresponding author. The data are not publicly available due to ethical approval of participant-informed consent which included survey respondents being informed that we would remove all personally identifiable information before sharing data with universities and/or research institutions.

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
