# Peer review of "UK Dog Owners’ Pre-Acquisition Information- and Advice-Seeking: A Mixed Methods Study"

_animals, 2024, doi:10.3390/ani14071033_

Round 1

Reviewer 1 Report

Comments and Suggestions for Authors

Following the authors’ previous paper published by Animals in 2023, this manuscript focused on further aspects of the information seeking by prospective owners prior acquiring a dog. Authors explored where dog owners looked for information, what type of information they sought, how long they spend on research and whether they found the information they wanted. By having a large sample size and a comprehensive survey/ semi-structured interview, this study enhances our understanding on how dog owners choose their dogs and how this process can be improved to benefit the human-animal relationship. Thus, it is a very interesting and important piece of research.

General aspects

Overall, the article is well organised and clear with sound methodology.

The review is relevant for the topic with appropriate references, showing that there is a lack of knowledge on how prospective dog owners look for information before acquiring a dog and thus, how to provide adequate and evidence-based information/advice to these population.

Specific comments

Title. It is a little confusing. Maybe reword it or delete “and advice”?

L46 to 51. It seems to me that these two phrases are out of place since the narrative seems that started again on line 49 “Dogs are the most popular companion animal species in the UK today”. I suggest rearranging it.

L72 to 75. This phrase is confusing. Maybe it could be divided into two? Also, it is not very clear the difference between the “quality of pre-acquisition research" and “quality of information” from line 81. I suggest to briefly explain what the former means.

L83 to 99. Although the information from these phrases is very interesting, I am not sure this paragraph needs to be so long. It gives the impression that these aspects were analysed by the present study. Maybe only cite the main findings? Lines 99 to 102, however, are important since their information leads to one of the objectives.

2 Materials and Methods. There seems to be many phrases/paragraphs the same as to the authors’ previous work (Mead et al., 2023). I appreciate that the methodology was very similar to your previous paper and it is quite challenging to rewrite the same information.

L132 to 132. It took me a while to distinguish “prospective” from “potential” owners, and within potential owners to differentiate between who already undertook research and who were planning to do so. Thus, I suggest making it clearer by adding this information in a table or bullet points (e.g., owners were grouped into three types, or definitions used).

2.2.3 Interviews. I think it would be useful to specify that the participants were different from the survey participants.

L271 to 273. I appreciate the detailed information, but if the variable “number of sources used by prospective owners” is not normally distributed, means do not need to be reported.

L284 to 286. This study managed to gather loads of important information, which is brilliant, and providing it at the supplementary materials is very helpful. My only suggestion here is that it doesn’t look like you ran any statistical analysis to identify differences within these demographic features between sources of information. Hence, you could cite that there is this material available (but not say there are differences). Likewise, the heading related to these tables at the Supplementary material gives the impression that significative differences were found (i.e., statistical analysis was carried out), thus I suggest changing it by something related to demographic aspects instead of “factors that influence…”.

Tables 1 and 2. They are very informative and providing the effect size is helpful. However, it is not clear which groups of owners differ in relation to each source of information. One way to address it would be to add different letters as superscript to the percentage or CI (e.g., if “Current owners” differ from “Potential owners (used)” but not from “Potential owners (plan to use)”, and no difference between “Potential owners” groups= X%a, Y%b and Z%a).

L303. “…especially where they were professionals” in the field?

3.3.1. Information about dogs: This section is a little confusing, since I had to read a few times and refer to the supplementary material to understand which were the themes and sub-themes. On the other hand, L378 to 382 were much clearer to me because the sub-themes were stated first and then you briefly explained and cited quotes of some of these sub-themes. Therefore, I suggest doing the same as L378 to 382, or maybe using a mind map (or other type of diagram) to show themes and their sub-themes.

Congratulations and good luck!

Author Response

Reviewer 1

Thank you for your time and effort in providing feedback on our manuscript. We have provided responses to all comments and suggestions in our point-by-point response (in blue) below.

Following the authors’ previous paper published by Animals in 2023, this manuscript focused on further aspects of the information seeking by prospective owners prior acquiring a dog. Authors explored where dog owners looked for information, what type of information they sought, how long they spend on research and whether they found the information they wanted. By having a large sample size and a comprehensive survey/ semi-structured interview, this study enhances our understanding on how dog owners choose their dogs and how this process can be improved to benefit the human-animal relationship. Thus, it is a very interesting and important piece of research.

Thank you for your kind comments about our research.

General aspects

Overall, the article is well organised and clear with sound methodology.

The review is relevant for the topic with appropriate references, showing that there is a lack of knowledge on how prospective dog owners look for information before acquiring a dog and thus, how to provide adequate and evidence-based information/advice to these population.

Specific comments

Title. It is a little confusing. Maybe reword it or delete “and advice”?

Thank you for your suggestion. We have discussed the title and on reflection feel that it is an appropriate description of this research. However, we defer to the editor and will alter the title if requested to do so.

L46 to 51. It seems to me that these two phrases are out of place since the narrative seems that started again on line 49 “Dogs are the most popular companion animal species in the UK today”. I suggest rearranging it.

Thank you for this suggestion. These lines have now been rearranged with the sentence “Dogs are the most popular companion animal species in the UK today” now being first and additional phrasing added to ensure the narrative reads better:

“Dogs are the most popular companion animal species in the UK today [1] with many households aspiring to acquire dogs each year. Decisions made during the dog acquisition process can have widespread implications for canine welfare [2-5] but despite this, there is currently limited knowledge about how prospective dog owners decide how and where to find their dogs, as well as which dogs to acquire. Whilst large numbers of dogs are acquired every year, tens if not hundreds of thousands are relinquished to welfare organisations each year [6,7].” (lines 46-53)

L72 to 75. This phrase is confusing. Maybe it could be divided into two? Also, it is not very clear the difference between the “quality of pre-acquisition research" and “quality of information” from line 81. I suggest to briefly explain what the former means.

Now lines 82-85 and 93-94. Thank you for highlighting this. We have done as you suggest and split this sentence into two and included a description of what is meant by “quality of pre-acquisition research”:

There is limited information on the quality of pre-acquisition research (i.e. the calibre of the research that prospective owners undertake prior to acquiring a dog). Neither is there much research into the time prospective owners spend on research […] (lines 82-85)

We have also described what is meant by “quality of information”:

Little is known about the quality of information from these sources (i.e. the standard or correctness of the information provided). (lines 93-94)

L83 to 99. Although the information from these phrases is very interesting, I am not sure this paragraph needs to be so long. It gives the impression that these aspects were analysed by the present study. Maybe only cite the main findings? Lines 99 to 102, however, are important since their information leads to one of the objectives.

Now lines 95-106 (and 108 & 112-115). Thank you for this suggestion. We appreciate that such level of detail is not needed and have now condensed this section.

2 Materials and Methods. There seems to be many phrases/paragraphs the same as to the authors’ previous work (Mead et al., 2023). I appreciate that the methodology was very similar to your previous paper and it is quite challenging to rewrite the same information.

Thank you for reading our previous work and noting that there are similarities. As you acknowledge, this paper reports on data that were collected as part of the same project and using the same methodology, necessitating repetition of some ideas in the earlier work. However, this current manuscript presents analysis that has been conducted and written separately as a stand-alone paper and all texts have been written based on the needs of this paper. No parts have been copied or repeated.

L132 to 132. It took me a while to distinguish “prospective” from “potential” owners, and within potential owners to differentiate between who already undertook research and who were planning to do so. Thus, I suggest making it clearer by adding this information in a table or bullet points (e.g., owners were grouped into three types, or definitions used).

Now lines ~144-147. Thank you for your comment. We appreciate that this could be confusing. Within our description and related analysis (e.g. Table 1), we have three categories:

  • Current owners (who undertook research)
  • Potential owners (who have undertaken research)
  • Potential owners (who plan to undertake research)

We hope that these three categories are clear, e.g. in Table 1 and by considering the relevant survey questions (described in the methods, lines 150-155, and in the SM).

The term “prospective owners” is used more generally to mean any of these owners at the stage before they have acquired their current dog, hence can be used to describe findings related to all respondents, particularly in section 3.3, without the need to specify the exact circumstances each time. Note that “prospective owners” are therefore not referred to in this section of the methods.

2.2.3 Interviews. I think it would be useful to specify that the participants were different from the survey participants.

Thank you for noticing this. As you are aware, we do refer to data collection being described in our previous paper (line 124); however, we should have provided details about how interviewees were recruited within this paper (some were recruited from survey participants). We have added wording to better explain this:

“Interviewees were recruited through the survey (n = 15), a pilot survey (n = 5), and pilot interviews with members of Dogs Trust staff (n = 12).” (lines 180-182)

L271 to 273. I appreciate the detailed information, but if the variable “number of sources used by prospective owners” is not normally distributed, means do not need to be reported.

Now lines 287-289. Thank you for this. These have now been removed.

L284 to 286. This study managed to gather loads of important information, which is brilliant, and providing it at the supplementary materials is very helpful. My only suggestion here is that it doesn’t look like you ran any statistical analysis to identify differences within these demographic features between sources of information. Hence, you could cite that there is this material available (but not say there are differences). Likewise, the heading related to these tables at the Supplementary material gives the impression that significative differences were found (i.e., statistical analysis was carried out), thus I suggest changing it by something related to demographic aspects instead of “factors that influence…”.

Now lines 300-302. Thank you for taking the time to look at our SM. We agree with your comments and have renamed this SM section accordingly to “Characteristics of prospective owners and sources of information utilised”.

Tables 1 and 2. They are very informative and providing the effect size is helpful. However, it is not clear which groups of owners differ in relation to each source of information. One way to address it would be to add different letters as superscript to the percentage or CI (e.g., if “Current owners” differ from “Potential owners (used)” but not from “Potential owners (plan to use)”, and no difference between “Potential owners” groups= X%a, Y%b and Z%a).

Thank you for this great suggestion. Both tables have been updated accordingly and wording included to explain this (lines 307-308 [Table 1] and lines 574-575 [Table 2]).

L303. “…especially where they were professionals” in the field?

Now line 320. Thank you for this. We have added your suggested wording to the end of this sentence.

3.3.1. Information about dogs: This section is a little confusing, since I had to read a few times and refer to the supplementary material to understand which were the themes and sub-themes. On the other hand, L378 to 382 were much clearer to me because the sub-themes were stated first and then you briefly explained and cited quotes of some of these sub-themes. Therefore, I suggest doing the same as L378 to 382, or maybe using a mind map (or other type of diagram) to show themes and their sub-themes.

Thank you for your comments. Section 3.3.1 has now been rearranged so that the sub-themes are introduced first, as suggested. What was previously the first sentence in this section has now been moved to the end (now lines 373-375).

Congratulations and good luck!

Thank you very much for taking the time to read our paper so thoroughly and providing such thoughtful comments. They are much appreciated and have helped us to improve this paper – thank you!

Reviewer 2 Report

Comments and Suggestions for Authors

The on-line survey is large and robust. The qualitative interview segment was very small, with only 24 current and 8 potential owners surveyed. This should be mentioned in the “strength and limitations” section, which is otherwise good at noting the potential limitations of the study. The exclusion of owners and potential owners who answered that they do not plan on seeking information does not impact the value of the study, but it would be worth examining the reasons for not seeking such information - i.e. past history of dogs in general or specific breed of interest, lack of access to resources, etc. It is unclear how large this number of excluded surveys was - is it the same as the values in Table 1 for “none of the above” and/or “Haven’t thought about it”?

Table 1 should also identify “Rescue Center staff/adoption counselors” as a source of information or advice. This is a significant source in many US animal shelters.

The key areas of information sought should b enumerated in a chart by % of respondents indicating such interest - particularly in view of increasing concerns about health concerns and breed-specific congenital disorders. If there is a comparatively low concern about such issues, this would help inform public information campaigns from veterinary and animal welfare groups.

As noted in the discussion, the degree of specificity about how much time prospective owners spent seeking  information or advice is likely relatively meaningless. The narrative of this result is sufficiently informative thus Table 2 seems unnecessary. A table of areas of information sought would be much more useful.

The relatively low incidence of the use or planned use of veterinarians as a source of reliable information is worth greater discussion since it does relate to concerns the authors raise about the quality of information that is provided by many of the sources being used. This finding should be of interest to the veterinary profession since it suggests that they are not doing a good job of presenting themselves as reliable sources of quality information. This may, in part, be linked to the fact that behavior, welfare and breed issues are still not well-covered in much veterinary training.

The discussion of the possible over-reliance on dubious breed stereotypes is excellent, as is the commentary on how few people considered if what they offered would be suitable for a dog.

Author Response

Reviewer 2

Thank you for your time and effort in providing feedback on our manuscript. We have provided responses to all comments and suggestions in our point-by-point response (in blue) below.

The on-line survey is large and robust. The qualitative interview segment was very small, with only 24 current and 8 potential owners surveyed. This should be mentioned in the “strength and limitations” section, which is otherwise good at noting the potential limitations of the study.

Thank you for your positive comments about our survey and potential limitations. Within qualitative interview studies, seemingly low numbers are relatively common, but they are actually not low within a qual context. Indeed, 32 interviews is quite a large number for an interview study. However, the bulk of our qualitative analysis was based on free text responses from survey respondents (as described in 2.3.2 Qualitative data analysis, lines 204-216 and 244-256, with exact breakdown per survey item provided in SM Table S2). Thus our qualitative findings are based on over 6,000 owners/potential owners.

Regardless, we appreciate your point and have included the following sentence in the limitations:

“Although a large amount of data was used for qualitative analysis, only a relatively small proportion of this was obtained through interviews. Additional interviews might provide deeper understanding of pre-acquisition behaviours.” (Lines 743-746)

The exclusion of owners and potential owners who answered that they do not plan on seeking information does not impact the value of the study, but it would be worth examining the reasons for not seeking such information - i.e. past history of dogs in general or specific breed of interest, lack of access to resources, etc. It is unclear how large this number of excluded surveys was - is it the same as the values in Table 1 for “none of the above” and/or “Haven’t thought about it”?

Thank you – this is an excellent point and indeed we have already undertaken this research: it is published and referenced within this paper (lines 80, 124, 148 and 264, reference 23, Mead et al. 2023, (lines 850-852).

This current manuscript focuses solely on those survey respondents who had undertaken research. This is noted in 3.1. Survey results (lines 259-264).Therefore, they are not included in any analyses in this paper, including Table 1.

Table 1 should also identify “Rescue Center staff/adoption counselors” as a source of information or advice. This is a significant source in many US animal shelters.

Thank you, this is a really interesting suggestion. Within the UK context this is not something that was specifically identified during our initial search of literature or pilot, including through qualitative data collection (free text responses and interviews) – all of which informed questions and response items in this final survey. (Indeed, I don’t think “adoption counsellors” is a term used in the UK – although this certainly sounds an interesting role!) Thus, these were not provided as options in our survey. This is not something that we can alter retrospectively but is certainly something we can consider within future research. Thank you for highlighting this. We have added this to our limitations:

Although most closed survey questions included “other” free text boxes, it is possible that providing more response options would have elicited additional responses. For example, rescue centre staff or adoption advisors were not included as response options in the question related to where prospective owners sought information or advice. It is possible these sources were used but not recorded as they were not offered as answers in the survey. Future research may like to consider additional options such as these.” (lines 737-743)

The key areas of information sought should b enumerated in a chart by % of respondents indicating such interest - particularly in view of increasing concerns about health concerns and breed-specific congenital disorders. If there is a comparatively low concern about such issues, this would help inform public information campaigns from veterinary and animal welfare groups.

Thank you for this suggestion. As these qualitative data were collected and analysed within a qualitative framework, it would not be appropriate to quantify the number of respondents articulating each theme. Through these qual data collection and analysis methods, we aimed to gather insights to improve understanding of the nature of pre-acquisition research: what this meant to prospective owners who were interviewed or who responded to these survey items, and what this included for them (see lines 193-199). The methods we describe are commonly used within qualitative research. The emphasis on this methodology is to gain an understanding of underlying reasons, opinions, experiences and motivations that cannot be gained from quantitative methods. Questions related to these areas were not designed to produce quantitative results. Instead, themes have been identified which are detailed in section 3.3 (lines 346-529) and are shown in Table 8 in the SM. Future research could consider taking these key areas of information sought and asking about them as part of a closed-item survey question, thus allowing for quantitative data collection and production of comparative %s.

As noted in the discussion, the degree of specificity about how much time prospective owners spent seeking  information or advice is likely relatively meaningless. The narrative of this result is sufficiently informative thus Table 2 seems unnecessary. A table of areas of information sought would be much more useful.

We agree that time on pre-acquisition research is very difficult to measure accurately and that what this means will vary hugely between prospective owners. Regardless of the limitations of the data, we still feel that this is an important insight into the length of time that prospective owners might be open to receiving information, prior to acquiring a dog (as noted in lines 698-701). This could be useful for those who wish to provide information to, or develop interventions for, prospective owners. Given that other studies have also attempted to quantify this time – but presumably have encountered similar issues in understanding what exactly this means – we feel that it is important to share these results and their limitations, should other researchers wish to consider this aspect of pre-acquisition research.

As previously explained, a table of information sought can be found in the supplementary materials but providing a numerical breakdown of these is not in keeping with data collection methods.

The relatively low incidence of the use or planned use of veterinarians as a source of reliable information is worth greater discussion since it does relate to concerns the authors raise about the quality of information that is provided by many of the sources being used. This finding should be of interest to the veterinary profession since it suggests that they are not doing a good job of presenting themselves as reliable sources of quality information. This may, in part, be linked to the fact that behavior, welfare and breed issues are still not well-covered in much veterinary training.

Thank you for this suggestion. We have included additional narrative relevant to veterinary professionals throughout our manuscript.

Within the introduction we have included the following (including an additional related reference):

“This is an important area for many animal welfare organisations and professionals. Indeed, in a 2023 survey which asked veterinary professionals about their main welfare concerns, a third chose “lack of adequate pre-purchase education regarding suitable pet choice” as the issue they would choose to resolve tomorrow and nearly a quarter considered this issue would have one of the biggest health and welfare implications in 10 years’ time if it was not tackled sooner [19].” (lines 69-74)

We have added the following wording to the discussion:

“Despite this apparent trust in friends and family who were considered to have relevant expertise or experience, few prospective owners consulted professionals such as vets or behaviourists. We found that less than 14% of prospective owners had consulted a vet. Smaller proportions have been reported in previous studies, including a 2017 survey that found only 6% of owners reported taking advice from veterinary professionals prior to acquiring a dog [26] and a 2020 survey that found just 3% of owners had undertaken a free online consultation with a vet prior to getting a pet [25]. Speculatively, Kuhl et al. [21] suggest that owners may not feel comfortable consulting with vets before acquiring an animal, may not be aware that veterinary professionals offer this support, or may be concerned about costs for such services. Prospective owners may also be driven by convenience consumer behaviour, leading to preference for seeking information that may be immediately accessible online. Despite this, there is clearly some interest in pre-purchase support from veterinary practices: within the same 2020 study, 45% of surveyed owners indicated that they would be interested in a free online consultation with a vet before acquiring a pet [25]. Indeed, nearly 30% of potential owners in our study reported that they were planning to seek advice from a vet prior to acquiring a dog. Pre-purchase consultations are championed by welfare organisations such as the UK veterinary charity, the People’s Dispensary for Sick Animals (PDSA) [42]. However, in the 2018 PDSA Animal Welfare (PAW) report, data collected from the British Veterinary Association (BVA) and British Veterinary Nursing Association (BVNA) reported that only 13% of veterinary practices offered free, dedicated pre-purchase clinics or appointments [43]. Lack of time, limited staffing, the need for practices to be profitable, and perceived public distrust in the veterinary profession have been identified as barriers to veterinary clinics engaging in pre-purchase consultations regarding brachycephalic dogs, but clearly more general research is needed into this area [44]. This might suggest that more could be done to promote professionals, such as veterinary professionals, as sources of quality information.(lines 610-635)

And:

“Dog professionals, such as vets, may wish to consider if and how they could better market themselves as offering pre-acquisition consultations to prospective owners.” (lines 721-723)

And to the conclusion:

“However, dog professionals such as vets, are less commonly consulted with regards to pre-acquisition advice.” (lines 754-755)

The discussion of the possible over-reliance on dubious breed stereotypes is excellent, as is the commentary on how few people considered if what they offered would be suitable for a dog.

Thank you very much for your comments.

Reviewer 3 Report

Comments and Suggestions for Authors

Dear authors,

Thank you for the opportunity to revise your manuscript. This is an interesting and timely issue related to dog ownership.

Before considering the manuscript for publication, some issues could be improved.

1. What was the main motivation for the research problem? Please, specify it in a clearer way in the introduction part.

2. contextualize your research with other similar studies.

3. Improve the research empirical works - there are recent papers that cover similar research problems.

4. Give more information about the procedure of data collection.

5. What are the main implications for both research and practice?

Author Response

Reviewer 3

Thank you for your time and effort in providing feedback on our manuscript. We have provided responses to all comments and suggestions in our point-by-point response (in blue) below.

Thank you for the opportunity to revise your manuscript. This is an interesting and timely issue related to dog ownership.

Before considering the manuscript for publication, some issues could be improved.

  1. What was the main motivation for the research problem? Please, specify it in a clearer way in the introduction part.

Our motivation is detailed in the introduction:

Better understanding of pre-acquisition information-seeking is needed as part of efforts to inform development of interventions aimed at improving decisions made by potential dog owners.” (lines 74-76)

However, we have added the following (including an additional related reference) prior to the above sentence:

“This is an important area for many animal welfare organisations and professionals. Indeed, in a 2023 survey which asked veterinary professionals about their main welfare concerns, a third chose “lack of adequate pre-purchase education regarding suitable pet choice” as the issue they would choose to resolve tomorrow and nearly a quarter considered this issue would have one of the biggest health and welfare implications in 10 years’ time if it was not tackled sooner [18].” (lines 69-74)

  1. contextualize your research with other similar studies.

Thank you for this. Within the Introduction (lines 77-115) we provide background information on similar areas of research. Similarly, we discuss our research within the context of other studies in the Discussion (lines 599-604 and 701-706).

Since submitting this paper for review, two relevant papers have been published: one reports on expectations of dog ownership (added ref 12, lines 54-58); the other reports on sources of information owners consulted prior to acquisition (added ref 26). Findings from the first paper are summarised as:

“For example, in a 2021 survey of UK dog owners, 52% reported that the cost of vet visits was higher than expected [12]. In the same study, many respondents described how they were surprised by the amount of resources, including time and money, that were required to fulfil a dog’s needs and meet the demands of dog ownership.” (lines 54-58)

Findings from the latter paper have been included within lines 87-92 and are discussed further within the context of our findings in lines 613-615 (additionally, this p/g has been extended following comments from another reviewer, hence lines 610-635 contain further information and discussion that help to contextualise our research, and include additional references).

We have also further discussed our findings within the context of similar research related to finding all the information sought / satisfaction with online information:

“In our study, considerably more respondents appeared satisfied with the information or advice they had found than did those surveyed by Kogan et al.[28] Their study found that over 50% of pet owners often felt “frustrated by a lack of information or an inability to find what I was looking for”, “sometimes”, “often” or “almost every time”. Although the two studies are not directly comparable as Kogan et al.’s study investigated the use of the internet only to source health information for current pet (not just dog) owners and questions asked were different, these contrasting findings suggest an area worthy of further exploration.” (lines 682-689)

However, given this is an under researched area of dog ownership, there are not many studies which our research can be compared to. Indeed, we consider our findings related to what information prospective owners seek in particular to be novel: we are not aware of any other studies regarding this area of preparation for dog ownership.

  1. Improve the research empirical works - there are recent papers that cover similar research problems.

As per our response to your point 2, we have added relevant findings from published papers.

  1. Give more information about the procedure of data collection.

We have provided considerable detail about data collection (2.2 Data collection, lines 135-183). We have also referenced previously published papers that provide more information about data collection for the whole project (lines 124-126).

  1. What are the main implications for both research and practice?

Thank you for your suggestion. We have added the following to the discussion to better consider implications:

“This study provides novel insights into the information that prospective owners seek and also offers insights into how this information is sought and the timescales over which this research is undertaken. Findings have implications for organisations and professionals involved in canine welfare and the provision of pre-acquisition advice. These stakeholders may want to think about: whether their advice includes the topics highlighted as of interest to prospective owners within this study; the quality of any resources; and the means by which they provide information. Dog professionals, such as vets, may wish to consider if and how they could better market themselves as offering pre-acquisition consultations to support prospective owners. (Lines 715-723)

Round 2

Reviewer 3 Report

Comments and Suggestions for Authors

Dear authors,

We appreciate your effort to address all the former comments. Therefore, I recommend the acceptance of the paper.